# A Functionalized Polysaccharide from *Sphingomonas* sp. HL-1 for High-Performance Flocculation

**DOI:** 10.3390/polym15010056

**Published:** 2022-12-23

**Authors:** Haolin Huang, Jingsong Li, Weiyi Tao, Shuang Li

**Affiliations:** 1College of Biotechnology and Pharmaceutical Engineering, Nanjing Tech University, Nanjing 211816, China; 2College of Food Science and Light Industry, Nanjing Tech University, Nanjing 211816, China

**Keywords:** bio-flocculant, *Sphingomonas*, sphingans, polysaccharide, wastewater treatment

## Abstract

The characterization and flocculation mechanism of a biopolymer flocculant produced by *Sphingomonas* sp. HL-1, were investigated. The bio-flocculant HL1 was identified as an acidic polysaccharide, mainly composed of glucose, and also contained a small amount of mannose, galacturonic acid and guluronic acid. The flocculating activity of the purified HL1 polysaccharide could be activated by trivalent cations, and its flocculation mechanism was mainly charge neutralization and bridging. The working concentration of fermentation broth HL1 in a kaolin suspension was only 1/10,000 (*v*/*v*), in which the polysaccharide concentration was about 2 mg/L. The bio-flocculant HL1 maintained high efficiency at a wide range of pH (pH 3–10). It also exhibited good flocculating activity at a temperature range of 20–40 °C; it could even tolerate high salinity and kept activity at a mineralization degree of 50,000 mg/L. Therefore, the bio-flocculant HL1 has a good application prospect in the treatment of wastewater over a broad pH range and in high salinity.

## 1. Introduction

The scientific and effective treatment of industrial wastewater is important because of the increasing discharge of wastewater and the accelerating process of industrialization [1]. Now, the most used wastewater treatment methods are efficient and environmentally friendly. These methods include coagulation, flocculation, precipitation and filtration [2]. Flocculation is widely used for its low cost, simplicity and effectiveness. Flocculants used in the flocculation process can be divided into inorganic flocculants, organic synthetic polymer flocculants and natural biological flocculants [3].

Microbial flocculant refers to a kind of organic matter with flocculation performance produced by microorganisms themselves or microbial fermentation, mainly through the interaction between microorganisms and their metabolites and particles in water to achieve a flocculation effect [4]. A polysaccharide bio-based flocculant is a kind of microbial flocculant; it has the advantages of a fast production speed, a high fermentation yield and an industrial application prospect. Compared with a traditional flocculant, it has the advantages of being a green, non-toxic flocculant, with no secondary pollution, easy degradation, sustainable regeneration and high flocculation efficiency [5].

Compared to proteins and polypeptide flocculants, the flocculating activity of polysaccharide bio-based flocculants are less affected by external factors. However, pH, temperature and metal ions still have some influence on its flocculating activity [6]. At the same time, they are related to the structure of polysaccharide itself and are also affected by dosage [7]. Many researchers believe that synthetic polymers have higher flocculating activity than natural polymers. Li et al. [8] synthesized an anionic polymer based on dextran, acrylamide and sodium acrylate by graft polymerization. When it was used as a flocculant, it only showed high flocculation performance in an alkaline environment, so there were still some limitations. Wang et al. [9] synthesized a cationic dextran-based flocculant, which showed high efficiency under acidic, neutral and alkaline conditions and can be degraded by enzymatic reactions easily. These kinds of biopolymers require high investment costs for equipment, energy consumption, production and purification. Therefore, it is still necessary to develop affordable natural biopolymers with excellent performances.

In our study, a novel polysaccharide-based bio-flocculant (HL1) was produced by the microbe *Sphingomonas* sp. HL-1. The production and separation of HL1 and its flocculating activity, flocculation mechanism and applicable environment were shown in this work.

## 2. Methods

### 2.1. Production and Preparation of Bio-Flocculant HL1

The strain *Sphingomonas* sp. HL-1 used in this study was deposited at China Center for Type Culture Collection (Wuhan University, Hubei, China) (CCTCC NO:M2021162). The seed medium contained 20 g/L of glucose, 5 g/L of yeast extract, 2 g/L of K_2_HPO_4_·3H_2_O, 0.1 g/L of MgSO_4_·7H_2_O at a pH of 7.0–7.2. The fermentation medium was comprised of 40 g/L of sucrose, 5 g/L of yeast extract, 2 g/L of K_2_HPO_4_·3H_2_O, 0.1 g/L of MgSO_4_·7H_2_O. The initial pH of the medium was adjusted to 7.0–7.2.

*Sphingomonas* sp. HL-1 was first inoculated into a 250 mL volumetric flask containing 50 mL of seed medium and cultured continuously for 24 h. The seed culture (6%, *v*/*v*) was then inoculated into 50 mL of fermentation medium for 72 h. All cultivation was carried out at 30 °C and 200 rpm.

### 2.2. Extraction and Purification of Exopolysaccharide HL1

The fermentation broth HL1 (HL1-FB) was extracted and purified by alcohol precipitation, which has been commonly used in polysaccharide extraction [10]. Firstly, the fermentation broth was diluted 1:1 with distilled water and put into a 75 °C water bath for 30 min. Then, the processed fermentation liquid was centrifuged to remove the cells. After concentration, a volume of ethanol (95%) 2–3 times the volume of the solution was added, put in a refrigerator overnight (4 °C) and centrifuged at 8000 rpm for 20 min. The crude polysaccharide products were obtained. An amount of the crude product was taken, and after complete swelling in the 75 °C water bath, an equal volume of the Savage reagent (CHCl_3_:BuOH = 5:1, *v*/*v*) to separate and precipitate proteins was added. After the removal of the precipitate constituted mainly by proteins, the supernatant was placed in a 10,000 Da dialysis bag for dialysis. The dialyzed solution was freeze-dried in order to obtain the purified polysaccharide HL1 (HL1-PS).

### 2.3. Characterization of HL1-PS

#### 2.3.1. Chemical Analysis of HL1-PS

Total sugar and protein contents in HL1-PS were quantified by the H_2_SO_4_-phenol method [11] and the Bradford method [12], respectively.

#### 2.3.2. Monosaccharide Composition Analysis

About 10 mg of HL1-PS was weighed precisely and placed in an ampoule. Then, 10 mL of a 3 M TFA (trifluoroacetic acid) was added, and the mixture was kept at 120 °C for 3 h for complete hydrolysis. A certain amount of the acid hydrolysate solution was transferred into a tube and blow dried by nitrogen gas. Then, 5 mL of water was added to the tube and mixed by vortex. Next, 100 μL of the diluted solution was pipetted into another tube, and 900 μL of deionized water was added to it. The resulting mixture was centrifuged at 12,000 rpm for 5 min. The supernatant was characterized by IC (ionic chromatography) analysis. The sample was loaded on an ICS5000 system (ThermoFisher, Waltham, MA, USA) equipped with a DionexCarbopac^TM^PA20 column at 30 °C [13].

A mixture of 16 standard monosaccharides was prepared (fucose, rhamnose, arabinose, galactose, glucose, xylose, mannose, fructose, ribose, galacturonic acid, glucuronic acid, aminogalactose hydrochloride, glucosamine hydrochloride, N-acetyl-D-glucosamine, guluronic acid and mannuronic acid) into 10 mg/mL of standard solution. Each monosaccharide standard solution was precisely configured with 5 mg/L of gradient concentration standard as standard. According to the peak area of the standard and the sample, the concentration of the sample was calculated, and the monosaccharide molar ratio of the sample was obtained according to the mass of different monosaccharides.
(1)$$\frac{C\left(standard\right)}{A\left(standard\right)}=\frac{C\left(sample\right)}{A\left(sample\right)}$$

#### 2.3.3. Potential Analysis

The zeta potentials of the reaction mixtures were measured by Mastersizer 3000 (Nano ZSE, Malvern, UK).

### 2.4. Flocculating Activity Assay

The flocculating activity of HL1-PS was evaluated by measuring its flocculation rate in a kaolin suspension [14]. This was done by using a 100 mL cylinder, placed on a magnetic stirrer, in which 1 mL of HL1 solution and 1 mL of polyferric chloride solution (0.3 mM) were added to 48 mL of kaolin suspension (6 g/L). The mixture was first stirred at 250 rpm for 1 min, then stirred at 100 rpm for 4 min and, finally, kept still for 5 min. The supernatant liquid (upper 2 cm) was taken, and its absorbance at 550 nm was measured by using a spectrophotometer (N4S UV-1800). Control experiments were performed in the same manner using deionized water instead of a HL1 solution. The HL1 solution was obtained by diluting HL-FB, and the dilutions were based on the experimental requirements (10-fold, 50-fold, 100-fold, 150-fold and 200-fold), and the total volume of each dilution was generally 10 mL. The flocculating rate (FR) was calculated by the following equation:(2)$$\mathrm{FR}=\frac{B-A}{B}\times 100\%$$
where *B* represents the absorbance of the control kaolin suspension at 550 nm, and *A* represents the absorbance of the flocculated sample system at 550 nm.

## 3. Results and Discussion

### 3.1. Extraction and Purification of Polysaccharide HL1

The yield of HL1-PS was about 20–25 g/L. The total sugar, uronic acid and protein contents of HL1-PS are shown in Table 1. The results of a monosaccharide composition analysis (Appendix A) showed that HL1-PS contained glucose, mannose, galacturonic acid, guluronic acid, arabinose, galactose and glucosamine hydrochloride (Table 1). By taking the above results into consideration, it can be shown that HL1-PS is an acidic polysaccharide, mainly composed of glucose, and it also contains a small amount of mannose, galacturonic acid and guluronic acid and only a tiny amount of arabinose, galactose and glucosamine hydrochloride.

Generally, the extracellular polysaccharides produced by *Sphingomonas* strains are called sphingans [15]. The chemical composition of typical sphingans, such as gellan gum, welan gum and diutan gum, are mainly composed of D-glucose, D-glucuronic acid, L-mannose and L-rhamnose (Schmid et al., 2014). However, the composition of sphingan HL1-PS contained a small quantity of mannose and no rhamnose, indicating that the composition of sphingan HL1-PS was significantly different from the typical composition of sphingans.

### 3.2. Effect of Metal Ions on Flocculating Activity of HL1-PS

Metal ions have been reported to affect flocculation function, most commonly Ca^2+^, Mg^2+^ and Fe^3+^ [16]. Their presence will not only improve the flocculation efficiency of bio-flocculants but also play an inhibitory role (Table 2). HL1-PS was used as a bio-flocculant, and the final concentration of polysaccharide HL1 was maintained at 4 mg/L in the reaction system. Metal ions (Na^+^, Ca^2+^, Fe^3+^ and Al^3+^) were used at the final concentration of 0.3 mM, and deionized water was added in the control group. The results are shown in Figure 1; the monovalent metal ion (Na^+^) and the divalent metal ion (Ca^2+^) slightly promoted the flocculation activity of HL1-PS, but trivalent ions (Fe^3+^ and Al^3+^) could significantly activate the flocculation function of HL1-PS, and the flocculation rate could reach more than 90%. This phenomenon is consistent with many reports indicating that biological flocculants depend on cations [3].

The effects of metal ions on the flocculation performance are closely related to the flocculation mechanism. Charge neutralization is an important mechanism for the flocculation reaction. The addition of metal ions neutralizes the charges and enhances the flocculating activity [21]. Cations mainly reduce electrostatic repulsion by neutralizing the negative charges of flocculants and suspended solids, thus, increasing the adsorption of polysaccharide bio-based flocculants on particle surfaces [22]. Monovalent cations produce loose bond structures compared to divalent cations, resulting in a decrease in flocculant density and size and flocculant resistance to shear [23]. Multivalent cations can bridge with negative functional groups on the biopolymer chain [5]. Therefore, the enhanced flocculating activity of HL1-PS by Fe^3+^ and Al^3+^ is mainly attributed to the bridging effect of multivalent cations, which also indicates that HL1-PS is a cation-dependent bio-flocculant.

### 3.3. Flocculation Mechanism of HL1-PS towards Kaolin Clay

The study of flocculation mechanisms is of great significance for optimizing flocculation parameters and improving practical application efficiency. The possible flocculation mechanisms proposed so far include charge neutralization, divalent cation bridging (DCB) theory and Derjaguin, Landau, Verwey and Overbeek (DLVO) theory. The DLVO theory is a classical colloidal theory that describes charged particles as having a double layer of counterions surrounding the particle [24]. Charge neutralization and DCB theory are considered to be the most relevant mechanism for polysaccharide flocculation [5]. In order to prove the flocculation mechanism of the polysaccharide HL1, the flocculation rate and zeta potential of different kaolin mixture systems were investigated, and the results are shown in Figure 2.

The initial zeta potential of the kaolin suspension was −22.7 mV. The zeta potential of the kaolin mixture after the addition of HL1-PS with a final concentration of 4 mg/L was further reduced to −26.5 mV due to the anionic character of the polysaccharide HL1. By the addition of 0.1 mM FeCl_3_, the zeta potential of the system slightly increased to −23.7 mV, and the flocculation rate reached 96.55%. This indicated that the slight change in the zeta potential of the kaolin suspension caused by Fe^3+^ could activate the flocculating activity of HL1-PS. On this basis, the concentration of FeCl_3_ was increased to 0.3 mM, the zeta potential increased to −8.91 mV, and the flocculation rate also increased to 99.81%. The positive charge added at this time neutralized the remaining negative charge, thus, increasing the flocculation rate. In addition, the zeta potential increased to 39.6 mV after the addition of 1.5 mM FeCl_3_, but the flocculation rate decreased to 16.18%, indicating that the neutralization of excess positive charges occurred at this time. Then, the concentration of HL1-PS in the system was increased to 20 mg/L, the flocculation rate increased to 69.09%, and the zeta potential only increased to 41.7 mV.

The slight change in zeta potential confirms the bridging mechanism of bio-flocculants [25]. Most natural flocculants have both charge neutralization and bridging effects in the flocculation process [26]. Therefore, the experimental results suggest that charge neutralization and bridging are the main flocculation mechanisms of the polysaccharide HL1.

### 3.4. Flocculation Characteristics of Bio-Flocculant HL1

In order to reduce costs, the purified polysaccharide HL1 (HL1-PS) and fermentation broth (HL1-FB) were used as flocculants towards kaolin clay, and their flocculating activity were compared. As shown in Table 3, both HL1-PS and HL1-FB showed excellent flocculation efficiency at the final working concentration of 4 mg/L and 1/5000 (*v*/*v*), respectively. Although the protein content in HL1-FB was much higher than that in HL1-PS, the zeta potential of the two samples were very similar, and they had similar flocculation efficiency. This indicates that HL1-FB could be directly used without any purification as a bio-flocculant with less costs and almost similar performances compared to HL1-PS.

#### 3.4.1. Effect of Dosage on Flocculation Activity

The dosage of flocculants affects the adsorption and electrostatic forces on the surface of colloidal particles [27]. HL1-FB was diluted and used directly, without any purification, as a flocculant to investigate the flocculation rate of kaolin at different dosages. The results are shown in Figure 3; the corresponding dosage of HL1-FB diluted 10-fold, 50-fold, 100-fold, 150-fold and 200-fold amounted to 1/500, 1/2500, 1/5000, 1/7500 and 1/10,000 (*v*/*v*) at final working concentration, respectively. At dilutions less than 100-fold, the flocculation rates were more than 99% without significant differences. For dilutions between 100-fold and 200-fold, the flocculation rate significantly decreased from 98.63% to 92.10%, which is due to the decrease in adsorption sites and the adsorbent surface area. According to previous studies, a flocculant can be considered to have high flocculation efficiency when the flocculation rate reaches more than 90% [5,27,28]. When HL1-FB was added at the amount of 1/10,000 (*v*/*v*) to the kaolin suspension, the flocculation rate exceeded 90%. This proves that the bio-flocculant HL1 had a good flocculation efficiency and has significant application prospects. Therefore, from the point of view of flocculation efficiency and economic cost, we used diluted HL1-FB (200-fold) for all the next experiments.

#### 3.4.2. Effect of pH on Flocculating Activity

The pH value has a certain impact on most flocculants. Polysaccharide bio-based flocculants are affected by the degree of ionization and binding forms in the presence of pH changes due to their structural and compositional peculiarities, leading to a significant effect on flocculation [29]. The pH adaptation test was performed with diluted HL1-FB (200-fold), and the results are shown in Figure 4a. Within the range of pH 3–10, the flocculating activity of HL1-FB was maintained at a very high level, all of which were higher than 99%. However, when the pH was raised to 11–12, the flocculation efficiency was greatly reduced. This indicated that the HL1-FB bio-flocculant showed flocculation activity across a wide range of environmental pH but could not tolerate an extremely alkaline environment. The currently reported bio-flocculants have always had a narrow pH range and unfavorable flocculation efficiency [19,30]. Among them, the bio-flocculant MBF-6 produced by *K. pneumoniae* YZ-6 exhibited relatively good results; it maintained more than 80% of flocculation activity in the pH of 3–11 [31]. Compared with these bio-flocculants, HL1-FB shows flocculation activity across a wider range of pH environments.

#### 3.4.3. Effect of Temperature on Flocculating Activity

Generally, polysaccharide bio-based flocculants have good thermal stability, while protein bio-based flocculants are easily damaged due to the influence of heat, thus, losing their flocculating activity [32]. The thermal stability of HL1-FB and the optimal reaction temperature of HL1-FB were tested in this work.

Diluted HL1-FB (200-fold) was treated at different temperatures for 1 h and then cooled to room temperature. The flocculation efficiency of heat-treated HL1 was tested at room temperature and a concentration of 2 mg/L. As shown in Figure 4b, HL1-FB exhibited good thermal stability as a bio-flocculant in the treatment range of 20–100 °C, and all of the flocculation rates were kept at more than 95%.

HL1-FB exhibited high flocculation activity (over 97%) in a working temperature range of 20–40 °C. However, it lost flocculating activity rapidly under high temperature (60 °C) (Figure 4c). As a bio-flocculant, HL1-FB could not tolerate high temperatures as high as 60 °C. Actually, we speculate that the loss of flocculation activity had nothing to do with HL1-FB itself. It has been highlighted that the zeta potential of polymeric ferric chloride changes when the temperature increases, leading to a decrease in the flocculation rate [33].

#### 3.4.4. Effect of Salinity on Flocculating Activity of HL1-FB

Salinity has an impact on sewage treatment in coastal areas and river channels, mainly on the flocculation and sedimentation of viscous sediments [34]. In particular, food processing and petroleum industries are facing challenges in removing particles from saline wastewater [35]. In order to explore the suitability of HL1-FB for use in saline systems, the flocculation rates of HL1-FB were tested with kaolin suspensions with various salinity. A kaolin suspension without salinity was used as the blank control.

As shown in Figure 4d, the flocculation activity of HL1-FB decreased gradually with the increase in salinity. However, the flocculation rates of HL1-FB remained above 80% when salinity was less than 10,000 mg/L. The flocculation activity of HL1-FB decreased gradually with a further increase in salinity but still remained above 70% at the salinity of 50,000 mg/L. The loss of flocculation activity may be due to the high positive charge content in the solution, which increased the repulsive forces between suspended particles and reduced the sedimentation velocity [34]. However, the salinity of 50,000 mg/L was a relatively high-mineralization environment, indicating that HL1-FB could be used in saline wastewater.

### 3.5. Analysis of Performance, Cost and Applications

According to previous reports, the flocculation rates of microbial flocculants were about 75–95% [27], with a dosage within the range of 0.1–150 mg/L [5]. In our study, HL1-FB was directly used as a bio-flocculant, which means that the conventional separation and extraction steps were omitted, thus, reducing the costs. HL1-FB exhibited high flocculating activity, and the minimum dosage of HL1-FB in the kaolin suspension was 1/10,000 (*v*/*v*). Thus, HL1-FB has the advantages of high efficiency and low cost, making it significantly competitive.

In the process of fermentation, the cost of the bio-flocculant HL1-FB mainly involved two aspects: one was the raw material cost, and the other was the processing cost. The processing costs mainly included electricity, steam consumption, consumed raw materials and labor costs for the fermentation process. According to the actual prices, the processing cost per ton of fermentation broth was about 1500 CNY. The raw materials needed for the production of 1 ton of fermentation broth were mainly 40 kg of glucose, 5 kg of yeast powder and a small amount of inorganic salt, which means that the raw material cost of each ton of fermentation broth should not exceed 500 CNY. Therefore, the total production cost of HL1-FB was about 2000 CNY/ton, which was around 2 CNY/L. According to the optimal reaction conditions, 0.1 L of HL1-FB and 63 mL of polyferric chloride (0.3 mM) were needed for the treatment of 1 ton of wastewater. The price of polyferric chloride was about 500 CNY/ton, and combined with the cost of HL1-FB, the cost of treating each ton of wastewater was about 0.2315 CNY.

## 4. Conclusions

A cationic-dependent polysaccharide-based bio-flocculant produced by the microbe *Sphingomonas* sp. HL-1 was characterized in our study. The polysaccharide HL1 was a novel sphingan with little mannose and no rhamnose. The polysaccharide HL1 was a trivalent cation-dependent bio-flocculant with charge neutralization and bridging as its main flocculation mechanism. The fermentation broth HL1 could be directly used as an efficient bio-flocculant, and the dosage could be as low as 1/10,000 (*v*/*v*). The bio-flocculant HL1 could tolerate a wide range of pH and also had good performance in saline wastewater. The bio-flocculant HL1 has a broad application prospect. 

## Figures and Tables

**Figure 1 polymers-15-00056-f001:**
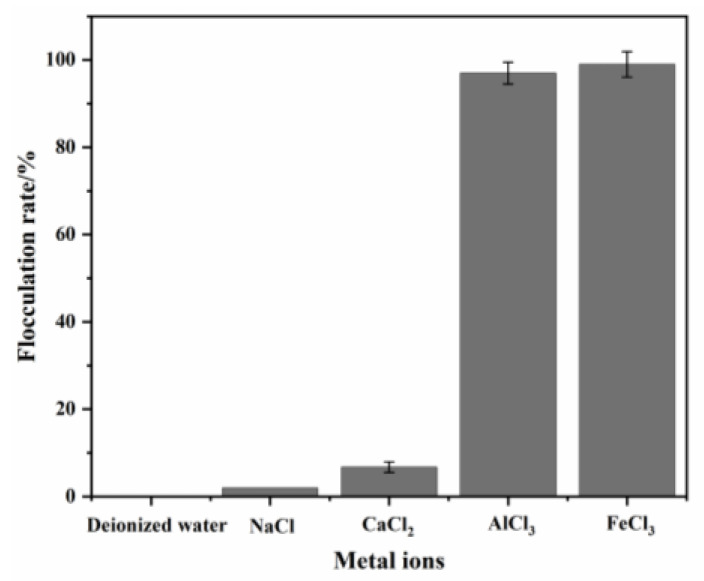
Effect of metal ions’ valency on the flocculating activity of HL1-PS.

**Figure 2 polymers-15-00056-f002:**
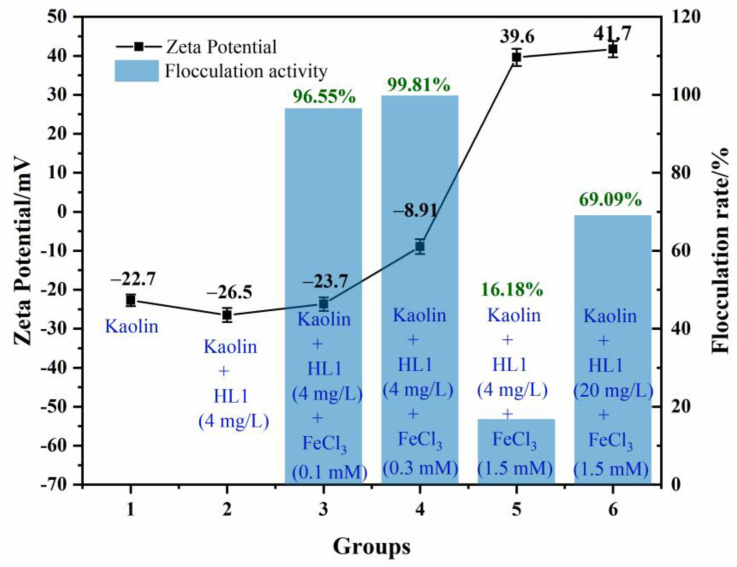
Zeta potential and flocculation activity of kaolin mixture system.

**Figure 3 polymers-15-00056-f003:**
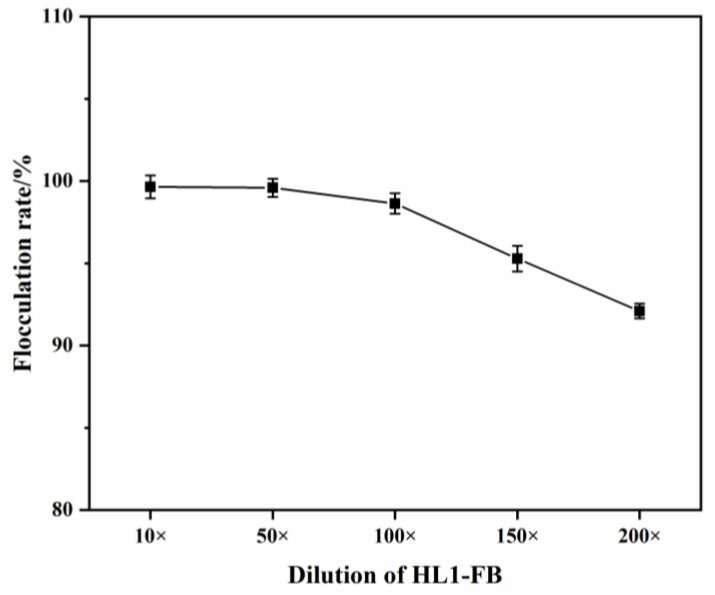
Flocculation rate of different dosage of HL1-FB.

**Figure 4 polymers-15-00056-f004:**
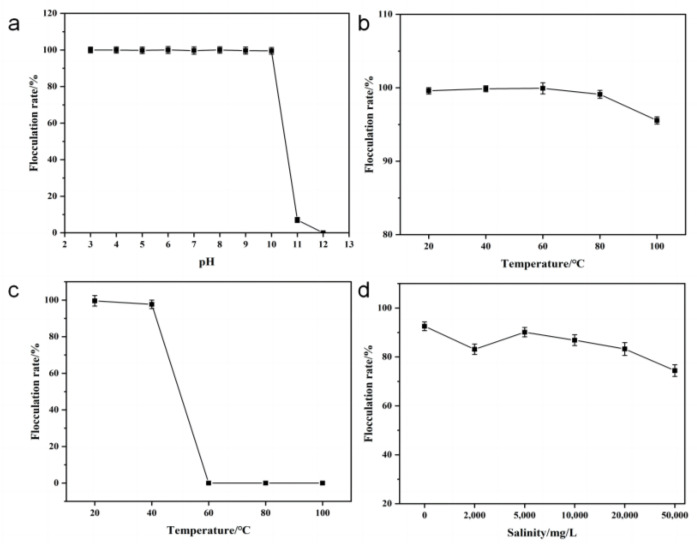
The effect of pH on the flocculating activity of HL1-FB (**a**); the thermal stability of HL1-FB (**b**); the working temperature of HL1-FB (**c**); and the effect of salinity on the flocculating activity of HL1-FB (**d**).

**Table 1 polymers-15-00056-t001:** Chemical composition of HL1-PS.

Component	Content
Total sugar (*w*/*w* %)	64.7
Uronic acid (*w*/*w* %)	14.2
Protein (*w*/*w* %)	4.6
Monosaccharide content (molar ratio, %)	
Glucose	89.3
Galacturonic acid	5.5
Guluronic acid	2.5
Mannose	1.9
Galactose	0.4
Arabinose	0.2
Glucosamine hydrochloride	0.2

**Table 2 polymers-15-00056-t002:** Effect of metal ion valency on the flocculating activity of various biological flocculants.

Strain	Promoted Metal Ions	Inhibited Metal Ions	Optimal FR	Ref.
*Bacillus toyonensis* AEMREG6	Na^+^, K^+^, Ca^2+^, Mg^2+^ and Al^3+^	Fe^3+^	87%	[17]
*Klebsiella* sp. ZZ-3	Fe^3+^ and Al^3+^	Monovalents (Na^+^ and K^+^) and divalents	94.5%	[18]
*Micrococcus* sp. Leo	K^+^, Ca^2+^, Mn^2+^, Ba^2+^, Fe^3+^ and Al^3+^	Na^+^, Li^+^ and Fe^3+^	85.2%	[19]
*Rhizopus* sp. M9 and M17	Mg^2+^ and Ca^2+^	Na^+^, K^+^, Al^3+^ and Fe^3+^	92.71%	[20]

**Table 3 polymers-15-00056-t003:** Flocculation rate of HL1-PS and HL1-FB.

Flocculant	Dosage	Zeta Potential	Fe^3+^ Content	FR	Zeta Potential
HL1-PS	4 mg/L	−26.5 mV	0.1 mM	96.55%	−23.7 mV
			0.3 mM	99.81%	−8.9 mV
HL1-FB	1/5000 (*v*/*v*)	−28.9 mV	0.1 mM	92.54%	−28.0 mV
			0.3 mM	99.59%	−19.4 mV

## Data Availability

Not applicable.

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
