# Peer review of "A Functionalized Polysaccharide from Sphingomonas sp. HL-1 for High-Performance Flocculation"

_polymers, 2022, doi:10.3390/polym15010056_

Round 1
Reviewer 1 Report
The paper is related to investigate the characterization and flocculation mechanism of a biopolymer flocculant produced by 9 Sphingomonas sp. HL-1. The idea is suitable for the journal, also, the manuscript is prepared very well and the English language of it is required very little changes ( in Introduction, methods and discussion sections) to show the main aims of the paper. Thus, I nominate this paper to publish in your respect journal
Author Response
- The paper is related to investigate the characterization and flocculation mechanism of a biopolymer flocculant produced by Sphingomonas sp. HL-1. The idea is suitable for the journal, also, the manuscript is prepared very well and the English language of it is required very little changes ( in Introduction, methods and discussion sections) to show the main aims of the paper. Thus, I nominate this paper to publish in your respect journal.
Thank you so much for your advice. We have made changes to the relevant content.

Reviewer 2 Report
Reviewer Report
Manuscript ID: Polymers 1998870
Title:
A Functionalized polysaccharide from Sphingomonas sp. HL1 for high performance flocculation.
Authors: Haolin Huang, Jinsong Li, Weiyi Tao and Shuang Li
This manuscript deals with the production, extraction and purification of exopolysaccharides using Sphingomonas sp. HL1. The monosaccharide composition of the resulting polysaccharides were determined quantitatively. The effect of the valence of metal ions (monovalent, divalent and trivalent metal ions) on the flocculation capacity of HL1 was studied. The flocculation mechanism was determined to include charge neutralization and divalent cation bridging.
The Zeta potential of a kaolin suspension was followed after adding the polysaccharide HL1 under different experimental conditions.
The flocculation capacity of the fermentation broth was determined also and it is possible the use the fermentation broth HL1 without any further purification.
The effect of dilution, pH, temperature and salinity on the flocculation capacity of the fermentation broth HL1 was assessed also.
Authors demonstrated that the exopolysaccharide HL1 can be used on a wide pH range (3-10) and in saline water within the temperature range 20-40°C.
General Comments:
Abstract:
Introduction:
Line 22-24: Please edit the following sentence:
“With the accelerating industrialization, the discharge of industrial wastewater is increasing day by day, which is very harmful to water resources, so it’s important to treat the wastewater effectively and scientifically [1].”
Methods:
2.3.2 Monosaccharide composition analysis
Line 84: Please add what IC stands for.
“Take the supernatant into IC analysis.”
2.4. Flocculating activity assay
Line 103: Please check the reference between parentheses. It is not necessary and not related to the magnetic stirrer. Please delete it:
“This was done by using a 100 ml cylinder, which was placed on a magnetic stirrer (N4S UV-1800),…”.
3. Results and Discussion:
Line 129: Please correct the following subtitles by changing “mental” by “metal” in the following subtitle:
“3.2. Effect of mental ions on flocculating activity of HL1”.
3.3. Flocculation mechanism of polysaccharide HL1 towards kaolin clay
Line 158-160: Please improve the following sentence:
“The possible flocculation mechanisms proposed so far include charge neutralization, Derjaguin, Landau, Verwey, and Overbeek (DLVO) theory and divalent cation bridging (DCB) theory [24].”
Punctuation should be check in the line 237.
Specific Comments:
I recommend authors to add the Chromatogram (Ionic Chromatography) of the purified HL1.
The chemical composition of the purified polysaccharide HL1 is given in table 1. Would you please specify which percentages are you used for the quantification of total sugar, uronic acid and proteins, w/w % or molar ratio%.
I recommend authors to add the exact composition of fermentation broth HL1 as it will be used later without any purification.
The protein content of the fermentation broth HL1 should be determined exactly as its flocculation performance will be affected with an increase in temperature.
I would appreciate if authors check the following sentences: Lines 232-241.
I find some inconsistency, as authors said “As shown in Fig 4b, HL1 exhibited good thermal stability as a 234 bio-flocculant in the treatment range of 20-100℃, and all of the flocculation rates kept more 235 than 95%.” And later they wrote “HL1 exhibited high flocculation activity (over 97%) in the working temperature 236 range of 20-40℃; However, it lost flocculating activity rapidly under high temperature 237 (60oC) (Fig 4c).”.
I found my self confused when authors use 1/5000 (v/v), then 1/10000 (v/v) and finally they dilute from 10 to 200 times. Can authors clarify more when they talk about diluting the fermentation broth.
I think authors are assessing the flocculation rate % of the purified HL1 in figure “b” and of the fermentation broth HL1 in figure “C”.
This part should be revised and discussed further based on the exact composition of the fermentation broth.
Authors should measure the Zeta potential of the fermentation broth in the same way as they did with the purified HL1, which will help them to discuss the results better.
I would appreciate if authors keep HL1 abbreviation when they talk about the purified exopolysaccharide HL1 and use another symbol when they talk about the fermentation broth HL1 in order to avoid any confusion.
Reference formatting should be checked, at least for the following references: 1, 5, 17, 32,
Decision:
For the above reasons, I think that this paper should be accepted only after Major revision.

Round 2
Reviewer 2 Report
Reviewer Report (Second Round)
Manuscript ID: Polymers 1998870
Title: A Functionalized polysaccharide from Sphingomonas sp. HL1 for high performance flocculation.
Authors: Haolin Huang, Jinsong Li, Weiyi Tao and Shuang Li
First, I would like to thank authors for the improvements they did to the manuscript.
But I still have some comments.
General Comments:
Lines 22-24:
Please change the following sentence:
“Efficiently, environmental friendly, commonly used wastewater treatment methods include coagulation, flocculation, precipitation and filtration [2].”
By the following sentence:
“Now, the most used wastewater treatment methods are efficient and environmental friendly. These methods include coagulation, flocculation, precipitation and filtration [2].”
Lines 25-28:
Please change the following sentence:
“The flocculation is widely used because of its low cost and convenient treatment. The flocculant used in the flocculation can be divided into inorganic flocculant, organic synthetic polymer flocculant and natural biological flocculant [3].”
by:
“Flocculation is widely used for its low cost, simplicity and effectiveness. Flocculants used in the flocculation process can be divided into inorganic flocculants, organic synthetic polymer flocculants and natural biological flocculants [3].”
Lines 37-39:
Please change the following sentence:
“Compared with protein and polypeptide flocculants, the flocculating activity of polysaccharide bio-based flocculants are less affected by external factors, but pH, temperature and metal ions still have some influence on its flocculating activity [6].”
by:
“Compared to proteins and polypeptide flocculants, the flocculating activity of polysaccharide bio-based flocculants are less affected by external factors. However, pH, temperature and metal ions still have some influence on its flocculating activity [6].”
Lines 45-48:
Please change the following sentence:
“Wang et al.[9] constructed MDT, a highly efficient cationic glucan based flocculant, showed high efficiency under acidic, neutral and alkaline conditions, and was easily degraded by enzymatic method, but it was expensive. Therefore, it is still necessary to develop natural biopolymers with excellent performance.”
By:
“Wang et al.[9] synthesized a cationic dextran based flocculant, which showed high efficiency under acidic, neutral and alkaline conditions, and can be degraded by enzymatic reactions easily. This kind of biopolymers requires high investment costs for equipment, energy consumption, production and purification. Therefore, it is still necessary to develop affordable natural biopolymers with excellent performances.”
Lines 55:
Please change the following sentence:
“Strain Sphingomonas sp. HL-1 used in study was deposited at China Center for Type 55 Culture Collection (CCTCC NO:M2021162).”
By
“Strain Sphingomonas sp. HL-1 used in this study was deposited at China Center for Type 55 Culture Collection (CCTCC NO:M2021162).”
Lines 71-75:
Please change the following sentence:
“Then take proper amount of crude products and swelling completely in 75℃ water bath, add equal volume Sevage reagent (CHCl3: BuOH = 5: 1, v/v) to remove the protein in the solution. The supernatant after protein removal was collected and placed in a 10000 Da dialysis bag for dialysis and freeze-drying to obtain purified polysaccharide HL1 (HL1-PS).”
By:
“Take an amount of the crude product and after complete swelling in the 75℃ water bath, add an equal volume of the Sevag reagent (CHCl3: BuOH = 5: 1, v/v) to separate and precipitate proteins. After removing the precipitate constituted mainly by proteins, the supernatant was placed in a 10000 Da dialysis bag for dialysis. The dialyzed solution was freeze-dried in order to obtain the purified polysaccharide HL1 (HL1-PS).”
Lines 81-87:
Please change the following sentence:
“The HL1-PS prepared by precision weighing 10mg was placed in ampoule, then 3M TFA (trifluoroacetic acid) 10mL was added and further hydrolyzed at 120℃ for 3h. A certain amount of acid hydrolysis solution was transferred to the tube for nitrogen blow drying, and 5ml water was added for vortex mixing. Add 900μL deionized water to 100μL solution, centrifuge at 12000 rpm for 5min. Take the supernatant into IC (ionic chromatography) analysis. Then sample was loaded on an ICS5000 system (Dionex, ThermoFisher, Amercian) equipped with a DionexCarbopacTMPA20 column at 30℃ [13].”
By:
“About 10 mg of HL1-PS was weighed precisely and placed in an ampoule. 10 mL of a 3M TFA (trifluoroacetic acid) was added and the mixture was kept at 120℃ for 3h for complete hydrolysis. A certain amount of the acid hydrolysate solution was transferred into a tube and blow dried by nitrogen gas. 5 ml of water was added to the tube and mixed by vortex. 100 μL of the diluted solution was pipetted into another tube and 900μL of deionized water was added to it. The resulting mixture was centrifuged at 12000 rpm for 5min. The supernatant was characterized by IC (ionic chromatography) analysis. The sample was loaded on an ICS5000 system (Dionex, ThermoFisher, Amercian) equipped with a DionexCarbopacTMPA20 column at 30℃ [13].”
Lines 88-97:
Please change the following sentence fragment:
“Take 16 kinds of monosaccharide standard (fucose, rhamnose, arabinose, galactose, glucose, xylose, mannose, fructose, ribose, galacturonic acid, glucuronic acid, galacturonic acid, galacturonic acid, glucosamine hydrochloride, glucosamine hydrochloride, N-acetyl-d-glucosamine, gulonuronic acid and mannose acid)…”.
By:
“A mixture of 16 standard monosaccharides was prepared (fucose, rhamnose, arabinose, galactose, glucose, xylose, mannose, fructose, ribose, galacturonic acid, glucuronic acid, glucosamine hydrochloride, glucosamine hydrochloride, N-acetyl-d-glucosamine, gulonuronic acid and mannose acid)…”
Lines 91-97:
Would you please clarify the meaning of the following sentence fragments and rewrite it, it is not clear:
“…into 10 mg/ml standard solution. Each monosaccharide standard solution was precisely configured with 0.01, 0.1, 0.5, 1, 5, 10, 20 mg/L gradient concentration standard as Standard 1-7. According to the absolute quantitative method, different monosaccharide mass was determined, and the molar ratio was calculated according to the molar mass of monosaccharide. The monosaccharide composition of HL1-PS could be obtained by comparing the retention time with that of the standard.”.
Lines 99-100:
Please change the following sentence:
“The zeta potential of reaction system were measured by Mastersizer 3000 (Nano ZSE, Malvern, UK).”.
By:
“The zeta potential of the reaction mixture was measured by Mastersizer 3000 (Nano ZSE, Malvern, UK).”.
Lines 102-108:
Please change the following sentence:
“The flocculating activity of HL1 was evaluated by measuring the flocculation rate of kaolin mixture simulation system [14]. This was done by using a 100 ml cylinder, which was placed on a magnetic stirrer, and 1mL diluted HL1 solution and 1ml diluted polyferric chloride solution (0.3 mM) were added to 48ml kaolin suspension (6 g/L). The mixture was first stirred at 250 rpm for 1 min, then stirred at 100 rpm for 4 min, and finally stayed still for 5 min. The upper 2 cm of liquid was taken and then measured the absorbance at 550 nm using a spectrophotometer (N4S UV-1800).”.
By:
“The flocculating activity of HL1-PS was evaluated by measuring its flocculation rate on a kaolin suspension [14]. This was done by using a 100 ml cylinder, placed on a magnetic stirrer, in which 1mL of HL1 solution and 1ml of polyferric chloride solution (0.3 mM) were added to 48ml kaolin suspension (6 g/L). The mixture was first stirred at 250 rpm for 1 min, then stirred at 100 rpm for 4 min, and finally stayed still for 5 min. The supernatant liquid (upper 2 cm) was taken and its absorbance at 550 nm was measured by using a spectrophotometer (N4S UV-1800).”.
Lines 146-147:
Please change the following word:
“The addition of metal ions neutralizes the charge…”
By:
“The addition of metal ions neutralizes the charges…”
Lines163-164:
Please change the following word:
“Charge neutralization and DCB theory are considered to be most relevant to the mechanism of polysaccharide flocculation [5].”.
By:
“Charge neutralization and DCB theory are considered to be the most relevant mechanism for the polysaccharide flocculation [5].”.
Page 4:
Please change the caption of Table 2 by:
“Table 2. Effect of metal ion valency on the flocculating activity of various biological flocculants.”
Please change the caption of Figure 1 by:
“Effect of metal ions’ valency on the flocculating activity of HL1-PS.”
Lines 168-181:
Please edit this paragraph and improve its quality:
“The initial zeta potential of kaolin suspension was -22.7 mV. Due to the anionic character of polysaccharide HL1, the zeta potential of kaolin mixture with 4 mg/L HL1-PS further reduced to -26.5 mV. By addition of 0.1 mM FeCl3, the zeta potential of the system slightly raised and reached about -23.7 mv, and significantly the flocculation rate of 96.55% was obtained. It indicated the slight change of zeta potential of kaolin suspension caused by Fe3+ could activate the flocculating activity of HL1-PS. On this basis, the concentration of FeCl3 was increased to 0.3 mM, the zeta potential was increased to -8.91 mV, and the flocculation rate was also increased to 99.81%. At this time, the addition of positive charge neutralizes the remaining negative charge, thus improving the flocculation rate. Furthermore, the zeta potential was increased to 39.6 mV, but the flocculation rate reduced to 16.18% by adding 1.5 mM FeCl3, indicating that neutralization of excess positive charge occurred at this time. Then, the concentration of HL1-PS in the system was increased to 20 mg/L, the flocculation rate was increased to 69.09% and the zeta potential only increased to 41.7 mV.”
Lines 182-186:
Please edit this paragraph and improve its quality:
“The slight change in zeta potential was shown to be evidence of a bridging mechanism occurring with the addition of bio-flocculants [25]. Most natural flocculants exhibited both charge neutralization and bridging mechanism during flocculation [26]. Therefore, the experimental results indicated that charge neutralization and bridging were the main flocculation mechanisms of polysaccharide HL1.”
Lines 190-197:
Please change the following sentence:
“In order to reduce the cost in practical application, the purified polysaccharide HL1 (HL1-PS) and fermentation broth (HL1-FB) were used as flocculant towards kaolin clay, respectively. As shown in Table 3, both the HL1-PS and HL1-FB showed excellent flocculation efficiency at the final working concentration of 4 mg/L and 1/5000(v/v), respectively. Although the protein content in HL1-FB was much higher than that in HL1-PS, the zeta potential of the two samples were very similar, and they had the similar flocculation efficiency. This indicated that the HL1-FB could be directly used as bio-flocculant with good cost performance.”
By:
“In order to reduce costs, the purified polysaccharide HL1 (HL1-PS) and fermentation broth (HL1-FB) were used as flocculants towards kaolin clay and their flocculating activity were compared. As shown in Table 3, both the HL1-PS and HL1-FB showed excellent flocculation efficiency at the final working concentration of 4 mg/L and 1/5000(v/v), respectively. Although the protein content in HL1-FB was much higher than that in HL1-PS, the zeta potential of the two samples were very similar, and they had similar flocculation efficiency. This indicated that the HL1-FB could be directly used without any purification as bio-flocculant with less costs and almost similar performances compared to the purified HL1-PS.”.
Lines 201-203:
Please change the following sentence:
“The dosage of flocculant affects the adsorption and electrostatic force on the surface of colloidal particles [27]. The HL1-FB was diluted and directly used as flocculant to investigate the flocculation rate of kaolin at different dosage.”
By:
“The dosage of flocculants affects the adsorption and electrostatic forces on the surface of colloidal particles [27]. The HL1-FB was diluted and used directly without any purification as flocculant to investigate the flocculation rate of kaolin at different dosages.”
Lines 206-214:
Please change the following sentence:
“At the dosage above 100-fold, the flocculation rates all reached more than 99% without significant difference. When the dosage varied from 100-fold to 200-fold, the flocculation rate was significantly increased from 92.10% to 98.63% due to the increase of adsorption site and adsorbent surface area. According to the previous studies, the flocculant can be considered to have high flocculation efficiency when the flocculation rate reaches more than 90% [5,27,28]. When the HL1-FB was added at the amount of 1/10000 (v/v) in the kaolin suspension, the flocculation rate had exceeded 90%. This proved that bio-flocculant HL1 had a good flocculation efficiency and significant application prospects.”
By:
“At dilutions less than 100-fold, flocculation rates were more than 99% without significant differences. For dilutions between 100-fold and 200-fold, the flocculation rate was significantly decreased from 98.63% to 92.10%, which is due to the decrease of adsorption sites and adsorbent surface area. According to the previous studies, A flocculant can be considered to have high flocculation efficiency when the flocculation rate reaches more than 90% [5,27,28]. When the HL1-FB was added at the amount of 1/10000 (v/v) to the kaolin suspension, the flocculation rate had exceeded 90%. This proved that bio-flocculant HL1 had a good flocculation efficiency and significant application prospects.”
Lines 218-232:
Would you please rewrite this paragraph and improve its quality:
“The pH value has a certain impact on most flocculants. Polysaccharide bio-based flocculants due to the particularity of structure and composition, in the case of pH change, the degree of ionization and binding form will be affected, leading to a significant impact on flocculation effect [29]. The diluted HL1-FB with 2 mg/L polysaccharide at working concentration were used to test the pH adaptability, and the result was shown in Fig 4a. Within the range of pH 3-10, the flocculating activity of HL1-FB was maintained at a very high level, all of which were higher than 99%. However, when the pH raised to 11-12, the flocculation efficient was greatly reduced. This indicated that HL1-FB bio-flocculant had a wide range of environmental pH, but could not tolerate extreme alkaline environment. At present, the pH range of bio-flocculants obtained are relatively narrow, and high flocculation efficiency and broad pH range are not easy to be achieved simultaneously [19,30]. Among them, the bio-flocculant MBF-6 produced by K. pneumoniae YZ-6 had a flocculation activity of more than 80% in the pH 3-11 range, and a maximum flocculation activity of 87% was observed at pH 7 [31]. Compared with these bio-flocculants, HL1-FB had a wider range of pH environments.”
Lines 254-269:
Please change the following sentence:
“Salinity has a certain impact on sewage treatment in coastal areas and river channels, mainly including flocculation and sedimentation of viscous sediments [34]. In particular, the food processing and petroleum industries are faced with the challenge of removing particles from saline wastewater [35]. In order to explore whether HL1-FB was suitable for application in saline system, the flocculation rates of HL1-FB were tested after the mixture of kaolin with various salinity. The kaolin suspension system without salinity was used as the blank control. As shown in Fig 4d, the flocculation activity of HL1-FB decreased gradually with increasing salinity. However, the flocculation rates of HL1-FB remained above 80% when the salinity less than 10000 mg/L. And then the flocculation activity of HL1-FB decreased gradually with the further increased salinity, but still remained above 70% at the salinity of 50000 mg/L. The lost of flocculation activity may be due to the high positive charge content in the solution, which increases the repulsive force between suspended particles and reduces the sedimentation velocity [34]. However, the salinity of 50000 mg/L was a relatively high mineralization environment, indicating that HL1-FB could be used in saline wastewater.”
By:
“Salinity has an impact on sewage treatment in coastal areas and river channels, mainly on the flocculation and sedimentation of viscous sediments [34]. In particular, food processing and petroleum industries are facing challenges in removing particles from saline wastewater [35]. In order to explore the suitability of HL1-FB to be used in saline systems, the flocculation rates of HL1-FB were tested with kaolin suspensions with various salinity. Kaolin suspension without salinity was used as the blank control. As shown in Fig 4d, the flocculation activity of HL1-FB decreased gradually with the increase of salinity. However, the flocculation rates of HL1-FB remained above 80% when salinity was less than 10000 mg/L. The flocculation activity of HL1-FB decreased gradually with further increase in the salinity, but still remained above 70% at the salinity of 50000 mg/L. The loss of flocculation activity may be due to the high positive charges content in the solution, which increased the repulsive forces between suspended particles and reduced the sedimentation velocity [34]. However, the salinity of 50000 mg/L was a relatively high mineralization environment, indicating that HL1-FB could be used in saline wastewater.”
Lines 271-289:
Please change the following sentence:
“According to previous reports, the flocculation rate of microbial flocculants was about 75%-95% [27], with the dosage ranging from 0.1-150 mg/L [5]. In our study, the HL1-FB could be directly used as bio-flocculant, which mean the conventional separation and extraction steps were omitted, greatly reducing the cost. HL1-FB exhibited high flocculating activity, and the minimum dosage of HL1-FB in kaolin suspension was 1/10000 (v/v). Thus, HL1-FB had the advantages of high efficiency and low cost, making it significantly competitive.
In the process of fermentation, the cost of bio-flocculant HL1-FB mainly involved two aspects, one was the raw material cost, the other was the processing cost. The processing cost mainly included the electricity and steam consumption, materials consumed and labor costs used in the fermentation process. According to the calculation, the processing cost per ton of fermentation broth was about 1500 CNY. The raw material of one ton of fermentation broth was mainly 40 kg glucose, 5 kg yeast powder and a small amount of inorganic salt, which mean that the raw material cost of each ton of fermentation broth should not exceed 500 CNY. Therefore, the total production cost of HL1-FB was about 2000 CNY/ton, which was 2 CNY /L. According to the optimal reaction conditions, 0.1L HL1-FB and 63 ml of polyferric chlorideis (0.3mM) were needed for per ton of wastewater. The price of polyferric chloride was about 500 CNY/ton, and combined with the cost of HL1-FB, the cost of treating each ton of wastewater was about 0.2315 CNY.”
By:
“According to previous reports, the flocculation rates of microbial flocculants was about 75%-95% [27], with a dosage within the range 0.1-150 mg/L [5]. In our study, the HL1-FB was directly used as a bio-flocculant, which means that the conventional separation and extraction steps were omitted, and thus reducing the costs. HL1-FB exhibited high flocculating activity, and the minimum dosage of HL1-FB in kaolin suspension was 1/10000 (v/v). Thus, HL1-FB had the advantages of high efficiency and low cost, making it significantly competitive.
In the process of fermentation, the costs of bio-flocculant HL1-FB mainly involved two aspects, one was the raw material costs, and the other was the processing costs. The processing costs included mainly electricity, steam consumption, consumed raw materials and labor costs for the fermentation process. According to the actual prices, the processing costs per ton of fermentation broth was about 1500 CNY. The raw materials needed for the production of 1 ton of fermentation broth was mainly 40 kg glucose, 5 kg yeast powder and a small amount of inorganic salt, which means that the raw material cost of each ton of fermentation broth should not exceed 500 CNY. Therefore, the total production cost of HL1-FB was about 2000 CNY/ton, which was around 2 CNY /L. According to the optimal reaction conditions, 0.1 L HL1-FB and 63 mL of polyferric chloride (0.3mM) were needed for the treatment of 1 ton of wastewater. The price of polyferric chloride was about 500 CNY/ton, and combined with the cost of HL1-FB, the cost of treating each ton of wastewater was about 0.2315 CNY.”
Specific comments:
Page 3:
In Table 1, why the sum of the chemical composition items of HL1-PS is not 100 % (total sugar, uronic acid and protein)?
Lines 203-206:
Would you please explain how you achieved the dilutions and include it in the paragraph 2.4. Flocculating activity assay.
Please indicate the total volume and the experimental procedure for the dilutions.
Page 6:
Would you please indicate by the end of the paragraph 3.4.1. Effect of dosage on flocculation activity, that you are going to use the lowest concentration of HL1-FB (diluted 200-fold) for the coming experiments.
Line 221:
“The diluted HL1-FB with 2 mg/L polysaccharide at working concentration were used to test the pH adaptability,…”.
I thought that authors used the concentration mg/L for the purified polysaccharide (HL1-PS) and the v/v concentration for the fermentation broth, non-purified polysaccharide, (HL1-FB)
Would you please explain or correct the above sentence.
Page 7:
3.4.3. Effect of temperature on flocculating activity:
What is the difference between Figure 4.b and Figure 4.c?
The same flocculant dosage, the same conditions but results are different. Why?
Decision:
For the above reasons, I think that this paper should be accepted only after Major Corrections (after making the above corrections).
Author Response
Thank you very much for your very careful and comprehensive comments on the linguistic aspects of our manuscript, which have been very helpful in improving it. We have revised most of the language of the article based on your comments and hope that this revision will meet your requirements.
- Page 3:
In Table 1, why the sum of the chemical composition items of HL1-PS is not 100 % (total sugar, uronic acid and protein)?
Generally speaking, on the one hand, due to the influence of the extraction method on the polysaccharides, a small portion of the sugars cannot be solubilized by water when the polysaccharide detection fraction is compounded, and on the other hand, due to the incompleteness of the detection method and the unknown substances present, resulting in the total sugar, glyoxylate and protein content not adding up to 100%. We have searched the literature to prove our claims, you can refer to “Free sugars and non-starch polysaccharides–phenolic acid complexes from bran, spent grain and sorghum seeds”.
- Lines 203-206:
Would you please explain how you achieved the dilutions and include it in the paragraph 2.4. Flocculating activity assay.
Please indicate the total volume and the experimental procedure for the dilutions.
Thank you so much for your advice. We use HL1-FB for dilution, generally by taking 1mL of HL1-FB and using the corresponding volumetric flask to fix the volume according to the dilution times (line 118-120).
- Page 6:
Would you please indicate by the end of the paragraph 3.4.1. Effect of dosage on flocculation activity, that you are going to use the lowest concentration of HL1-FB (diluted 200-fold) for the coming experiments.
As suggested by the reviewer, we added the concentration of HL1-FB (200x) used after 3.4.1 (line 224-226).
- Line 221:
“The diluted HL1-FB with 2 mg/L polysaccharide at working concentration were used to test the pH adaptability,…”.
I thought that authors used the concentration mg/L for the purified polysaccharide (HL1-PS) and the v/v concentration for the fermentation broth, non-purified polysaccharide, (HL1-FB)
Would you please explain or correct the above sentence.
We are sorry for that, we modified the concentration of the diluted HL1-FB (200-fold) (line 234).
- Page 7:
3.4.3. Effect of temperature on flocculating activity:
What is the difference between Figure 4.b and Figure 4.c?
The same flocculant dosage, the same conditions but results are different. Why?
We are very sorry for the confusion. The Fig 4b showed the thermal stability of HL1-FB, and the Fig 4c showed the optimal reaction temperature of HL1-FB. The relevant description is in 3.4.3 (line 248-251).

Round 3
Reviewer 2 Report
Reviewer Report (Third Round)
Manuscript ID: Polymers 1998870
Title:
A Functionalized polysaccharide from Sphingomonas sp. HL1 for high performance flocculation.
Authors: Haolin Huang, Jinsong Li, Weiyi Tao and Shuang Li
First, I would like to thank authors for the improvements they did to the manuscript.
I have few minor corrections.
Lines 25:
Please change the following sentence:
“Now, the most used wastewater treatment methods are efficient and environmental friendly...”
By:
“Now, the most used wastewater treatment methods are efficient and environmentally friendly...”
Lines 70-72:
Please change the following sentence:
“After concentration, 2-3 times the volume of 95 % ethanol was added, put into 4℃ refrigerator overnight and centrifuged at 8000 rpm for 20 min.”
By:
“After concentration, a volume of ethanol (95%) 2-3 times the volume of the solution was added, put in a refrigerator overnight (4℃) and centrifuged at 8000 rpm for 20 min.”
Line 89:
Please change the following sentence:
“Then sample was loaded on an ICS5000 system…”
By:
“The sample was loaded on an ICS5000 system…”
Line 102”
Please change the following sentence:
“The zeta potential of the reaction system were measured by Mastersizer 3000”
By:
“The zeta potential of the reaction mixtures were measured by Mastersizer 3000”
Lines 116-118:
I find the following sentence not clear, would you please improve it:
“The dilute them, 1mL HL-FB was added into a volumeter flask of corresponding volume and then the volume was fixed to obtain HL-FB diluent, while HL-PS diluent could be directly configured from the sample according to the required concentration.”
Lines 132-134:
Please change the following sentence:
“However, the component of sphingan HL1-PS contained only little mannose and no rhamnose, indicating that the structural component of sphingan HL1-PS was significantly different with typical sphingans.”
By:
“However, the composition of sphingan HL1-PS contained little quantity of mannose and no rhamnose, indicating that the composition of sphingan HL1-PS was significantly different from the typical composition of sphingans.”.
Line 141:
Please change:
“Metal ion (Na+, Ca2+, Fe3+ and Al3+) were…”.
By:
“Metal ions (Na+, Ca2+, Fe3+ and Al3+) were…”.
Lines 146-147:
Please change the following sentence:
“This phenomenon was consistent with many reports that biological flocculants depended on cations [3].”
By”
“This phenomenon was consistent with many reports indicating that biological flocculants depended on cations [3].”
Lines 174-175:
Please change the following sentence:
“The zeta potential of the kaolinite mixture with the addition of 4 mg/L HL1-PS was further reduced to -26.5 mV due to the anionic character of the polysaccharide HL1.”
By:
“The zeta potential of the kaolin mixture after addition of HL1-PS with a final concentration of 4 mg/L was further reduced to -26.5 mV due to the anionic character of the polysaccharide HL1.”.
Line 188:
Please change the following sentence fragment:
“The slight change in zeta potential demonstrate…”
By:
“The slight change in zeta potential confirms…”
Lines 233-237:
Would you please check and revise the following sentence:
“The currently obtained bioflocculants have a narrow pH range and do not easily achieve high flocculation efficiency and wide pH range simultaneously [19,30]. Among them, the bio-flocculant MBF-6 produced by K. pneumoniae YZ-6 had a flocculation activity of more than 80% in the pH 3-11 rangeand the highest flocculation activity of 87% at pH 7 [31].”
Lines 248-249:
Please change the following sentence:
“The flocculation efficiency of heat-treated HL1 were tested 248 at the dosage of 2 mg/L.”
By:
“The flocculation efficiency of heat-treated HL1 were tested at room temperature and a concentration of 2 mg/L.”
Lines 254-256:
Please change the following sentences:
“Superficially, HL1-FB as a bio-flocculant could not tolerate high ambient temperature of 60℃. Actually, we speculated the loss of flocculation activity had nothing to do with HL1-FB itself.”
By:
“As a bio-flocculant, HL1-FB could not tolerate high temperatures as high as 60℃. Actually, we speculate that the loss of flocculation activity had nothing to do with HL1-FB itself.”
Line 276:
Please change the following sentence:
“According to previous reports, the flocculation rates of microbial flocculants was about 75%-95%...”
By:
“According to previous reports, the flocculation rates of microbial flocculants were about 75%-95%...”
Decision:
For the above reasons, I think that this paper should be accepted only after Minor Corrections (after making the above corrections).

Author Response
Thank you very much for your very careful and comprehensive comments on the linguistic aspects of our manuscript, which were very helpful in improving our manuscript. We have made changes to the language based on your comments and hope that they will meet your requirements. The following are responses to the statements you would like to see improved.
- I find the following sentence not clear, would you please improve it:
“The dilute them, 1mL HL-FB was added into a volumeter flask of corresponding volume and then the volume was fixed to obtain HL-FB diluent, while HL-PS diluent could be directly configured from the sample according to the required concentration.”
As suggested by the reviewer, we illustrate the HL1-FB dilution multiples and the total volume, and modify the statements in line 114-117.
- Lines 233-237:
Would you please check and revise the following sentence:
“The currently obtained bioflocculants have a narrow pH range and do not easily achieve high flocculation efficiency and wide pH range simultaneously [19,30]. Among them, the bio-flocculant MBF-6 produced by K. pneumoniae YZ-6 had a flocculation activity of more than 80% in the pH 3-11 rangeand the highest flocculation activity of 87% at pH 7 [31].”
Thank you so much for your advice. We have revised the part of the sentence that is easily misunderstood (line 239-243).
